# A Fair Loss Function for Network Pruning

**Robbie Meyer**
University of Waterloo
Waterloo, Ontario, Canada
`robbie.meyer@uwaterloo.ca`

**Alexander Wong**
University of Waterloo
Waterloo, Ontario, Canada
`alexander.wong@uwaterloo.ca`

## Abstract

Model pruning can enable the deployment of neural networks in environments with resource constraints. While pruning may have a small effect on the overall performance of the model, it can exacerbate existing biases into the model such that subsets of samples see significantly degraded performance. In this paper, we introduce the performance weighted loss function, a simple modified cross-entropy loss function that can be used to limit the introduction of biases during pruning. Experiments using biased classifiers for facial classification and skin-lesion classification tasks demonstrate that the proposed method is a simple and effective tool that can enable existing pruning methods to be used in fairness sensitive contexts.

## 1   Introduction

Deep learning models are large, requiring millions of operations to make an inference [1]. Deploying large neural networks to environments with limited computational resources, such as mobile and embedded devices, may be infeasible.

Pruning is a simple and common method for reducing the size of a neural network [2]. It involves identifying parameters that do not significantly affect the model's output and removing them from the network. Pruning enables the deployment of performant neural networks in resource constrained environments [3, 4]. However, recent research has shown that while overall accuracy of the model may be maintained while the model is compressed, pruning can exacerbate existing model biases, disproportionately affecting disadvantaged groups [5]. Pruning methods that are designed to preserve overall model performance may not prioritize the preservation of parameters that are only important for a small subset of samples.

This effect has significant implications for the implementation of pruning in real-world situations. Biases have been observed in artificial intelligence systems such as those used to classify chest X-ray images [6], recognize faces [7] and screen resumes [8]. Biases in models can increase the risk of unfair outcomes, preventing the implementation of the model. If pruning exacerbated a model's biases, it could increase the risk of unfair outcomes or limit the deployment of the pruned model. It is therefore important to prune in a manner that does not aggravate a model's biases.

In this paper, we propose the performance weighted loss function as a simple method for boosting the fairness of data-driven methods for pruning convolutional filters in convolutional neural network image classifiers. The goal of our method is to enable the pruning of a significant number of model parameters without significantly exacerbating existing biases. The loss function consists of two small tweaks to the standard cross-entropy loss function to prioritize the model's performance for poorly-classified samples over well-classified samples. These tweaks can be used to extend existing data-driven pruning methods without requiring explicit attribute information.

We demonstrate the effectiveness of our approach by pruning classifiers using two different pruning approaches for the CelebA [9] and Fitzpatrick17k [10] datasets. Our results show that the performance

2022 Trustworthy and Socially Responsible Machine Learning (TSRML 2022) co-located with NeurIPS 2022.

weighted loss function can enable existing pruning methods to prune neural networks without significantly increasing model bias.

## 2 Related Work

Many different pruning approaches have been proposed to reduce the size of CNNs while minimally impacting model accuracy. Pruning methods typically involve assigning a score to each parameter or group of parameters, removing parameters based on these scores and retraining the newly pruned network to recover lost accuracy [2].

The procedure by which parameters are identified to be pruned is the primary differentiator between pruning methods. There are a wide variety of scoring approaches used to identify parameters that are unimportant or redundant and can be removed from the network. Many approaches use parameter magnitudes to identify parameters to prune [11, 12]. Other approaches use gradient information [13], Taylor estimates of parameter importance [14, 15, 16] and statistical properties of future layers [17]. Some approaches involve learning the scores via parameters that control the flow of information through the network [18, 19].

However, almost all novel pruning approaches focus on the overall accuracy of the model after pruning. There are few papers that aim to improve or analyze the fairness of a pruned model. Hooker et al. [5] propose auditing samples affected by model compression, called Compression Identified Exemplars, as an approach for identifying and managing the negative effects of model compression. Paganini [20] demonstrates how class fairness can be affected by pruning approaches that only consider overall accuracy. Wu et al. [21] propose Fairprune, a method for improving model bias using pruning. Instead of seeking to compress a model, Fairprune prunes parameters using a saliency metric to increase model fairness [21]. Xu and Hu [22] propose the use of knowledge distillation and pruning to reduce bias in natural language models. Joseph et al. [23] propose a multi-part loss function intended to improve the alignment between predictions between the original and pruned model. They demonstrate that their method can have beneficial effects for fairness between classes. Marcinkevičs et al. [24] propose a debiasing procedure that involves pruning parameters using a gradient based influence measure.

While not a pruning method, the work of Mahabadi and Henderson [25] is also relevant as their "Debiased Focal Loss" resembles the weighting scheme of the loss function proposed in this paper. Instead of using their loss function for model compression, they aim to debias a model using the output of a trained bias-only model.

## 3 Method

### 3.1 Motivation

In the unfair pruning situation described by Hooker et al. [5], model performance was more significantly impacted for certain sample subgroups. The highly impacted subgroups were characterized by poor representation in the training data and worse subgroup performance by the original model when compared to unimpacted groups. The performance decrement induced by the pruning process disproportionately impacts subgroups which are underrepresented and poorly classified.

To rectify this inequality, we can design a pruning process that prioritizes maintaining the performance of samples from the impacted subgroups. However, we do not need to develop a new pruning method from scratch to achieve this objective. Many existing pruning methods use data to identify which model parameters should be removed. Some methods use parameters learned via a loss minimization process whereas others values derived from gradients calculated with respect to a loss function. By modifying the loss function to prioritize samples from impacted subgroups, we can boost the fairness of existing pruning methods.

### 3.2 The Performance Weighted Loss Function

We make two different modifications to the standard cross-entropy loss function to transform it into the performance weighted loss function (PW loss). We first apply sample weighting to ensure that

samples from impacted groups have a larger contribution to the loss function. We then transform the sample labels to ensure that we are not reinforcing undesirable model behaviours.

As the attribute information required to identify impacted subgroups is not always readily accessible, our weighting scheme does not depend on any external information. We instead use the output of the original model to determine each sample weight. We assign larger weights to samples for which the original model was not able to confidently classify. The form of the scheme resembles the focal loss [26]. However, as the samples are weighted using the outputs of the original model the weights do not depend on the current output of the model and will not change during training. The weight assigned to the $i$th data sample, $w_i$, is given by the following equation:

$$w_i = \theta + (1 - \hat{y}_i)^\gamma \tag{1}$$

where $\hat{y}_i$ is the predicted probability given by the original model for the sample's true class, $\theta \in [0, 1]$ is the minimum weight value and $\gamma \geq 0$ controls the shape of the relation between $\hat{y}_i$ and $w_i$.

We also emphasize the model performance through the use of corrected soft-labels in the cross-entropy function. Rather than using the true labels of each sample, we use the output of the original model for the loss function in the pruning process. Without this change, the preservation of an originally poorly classified sample's prediction probability would result in a greater loss value than the preservation of an originally well classified sample's prediction probability. The use of true labels implicitly prioritizes the preservation of model performance for samples that have predictions closer to their true labels. Using the model output as soft-labels alleviates this implicit prioritization.

However, as we are assigning higher weights to samples that are originally classified by the original model while also using the original model's output as our labels, we are consequently assigning the highest weights to incorrect labels. To avoid emphasizing incorrect behaviours we correct the soft-labels. The corrected soft-label, $\hat{\boldsymbol{y}}_i^*$ is defined as:

$$\hat{\boldsymbol{y}}_i^* = \begin{cases} \hat{\boldsymbol{y}}_i & \text{if} \quad \hat{C}_i = C_i \\ \boldsymbol{y}_i & \text{otherwise} \end{cases} \tag{2}$$

where $\hat{\boldsymbol{y}}_i$ contains the prediction probabilities derived from the model output for the $i$th sample, $\boldsymbol{y}_i$ is the true label vector of the $i$th sample, $\hat{C}_i$ is predicted class of the $i$th sample and $C_i$ is the true class of the $i$th sample. The corrected soft-label takes on the value of the model's prediction probabilities when the prediction is correct and the true label when the prediction is incorrect.

By the application of the performance weighted scheme and corrected soft-labels onto the standard cross-entropy function, the performance weighted loss function, $\mathcal{L}_{PW}$, is defined by:

$$\mathcal{L}_{PW} = \sum_{i=1}^{N} w_i l_{CE}(\hat{\boldsymbol{y}}_i^*, \hat{\boldsymbol{y}}_i') \tag{3}$$

where $\hat{\boldsymbol{y}}_i'$ contains the prediction probabilities derived from the model output for the $i$th sample after pruning, $l_{CE}(\hat{\boldsymbol{y}}_i^*, \hat{\boldsymbol{y}}_i')$ is the cross-entropy between the corrected soft-label and the prediction probabilities of the pruned model for the $i$th sample, and $N$ is the number of samples in the batch.

By using this loss function with existing data-driven pruning methods, we can reduce the bias exaggerating effect of pruning by emphasizing samples that are more likely to be negatively affected by pruning.

## 4 Experiments

### 4.1 Experimental Set-up

We applied the PW loss to two different pruning methods. The first method is AutoBot [18], an accuracy preserving pruning method that uses trainable bottleneck parameters that limits the flow of information through the model. The second method uses an importance metric derived from the Taylor expansion of the loss function [14]. In both of our implementations, we pruned whole convolutional filters rather than individual neurons. As pruned filters can be fully removed from the model, rather than being set to zero, filter pruning is a simple method for directly reducing the FLOPS of a model.

In the AutoBot method, the bottlenecks are optimized by minimizing a loss function that includes the cross-entropy between the original and pruned model outputs, as well as terms that encourage the bottlenecks to limit information moving through the model, achieving a target number of FLOPS [18]. We applied the performance weighted loss function to the method by replacing the cross-entropy term in the loss function with the performance weighted loss function. Additionally, we also used the performance weighted loss function when retraining the model after pruning.

The importance metric of the Taylor expansion method is formed using the gradient of the loss function with respect to each feature map and the value of each feature map [14]. This method alternates between training the network and pruning a filter. In our implementation, a filter is pruned every five iterations. We applied the performance weighted loss function by replacing the loss functions used in the gradient calculation and model training with the performance weighted loss function. Once again, we also used the performance weighted loss function when retraining the model after pruning.

We also evaluated a random pruning method in which filters are selected and pruned from the network until only the desired number of FLOPS remain. We use this method as a reference.

We implemented the methods using the *PyTorch* library [27]. The methods were implemented as three step pipelines in which the model is first pseudo-pruned by setting parameters to zero, fully pruned using the *Torch-Pruning* library [28] and retrained. Pseudo-pruning allows for fast pruning during the pruning process while the full pruning step removes the unused parameters, reducing the number of operations required for prediction. Due to dependencies between parameters introduced by structures such as residual layers, the achieved theoretical speedup often slightly differs from the target theoretical speedup. All hyperparameters for the pruning methods were selected using a hold-out validation set. Hyperparameters for the pruning methods were selected without the PW loss applied and were used for both unmodified and PW loss method variants. We repeated each experiment three times. All figures displaying model performance after pruning are displaying the average of all trials.

### 4.1.1 Metrics

Our primary concern is the degradation of a model's behaviour towards different subgroups due to pruning. We therefore evaluated the models by comparing the change in the areas under the receiver operator curves (ROC-AUC) for various subgroups for five different degrees of pruning. As it is a threshold agnostic performance metric, the ROC-AUC is a good measure of the model's understanding and separability for a subgroup [29]. For non-binary classification we used the one-vs-one ROC-AUC.

We measured the degree to which a model is pruned using the theoretical speedup, defined as the FLOPS of the original model divided by the FLOPS of the pruned model.

### 4.2 Evaluating Fairness and Performance

All methods were tested with and without the PW loss on two different classification tasks.

Our first task was the celebrity face classification task using the CelebA dataset [9] as outlined by Hooker et al. [5], in which a model is trained to identify faces as blonde or non-blonde. The CelebA dataset contains over 200 000 images of celebrity faces with various annotations. While blonde non-male samples make up 14.05% of the training data, blond male samples make up only 0.85% of the training data. We used the provided data splits with 80% of the available data being used for training with the remaining data split evenly for validation and testing.

Our second task is the skin lesion classification task using the Fitzpatrick17k dataset [10]. The Fitzpatrick17k dataset consists of 16 577 images of skin conditions. We trained our models to classify the samples as non-neoplastic, benign or malignant. Due to missing and invalid images we were only able to use 16 526 images. Each sample in the dataset is assigned a Fitzpatrick score that categorizes the skin tone of the sample. We trained our models on only samples with light skin tone scores of 1 or 2, and evaluated the model on medium skin tone scores of 3 or 4 as well as dark skin tone scores of 5 or 6. We used a random 25% of the medium and dark skin tones as a validation set with the remainder used as a test set.

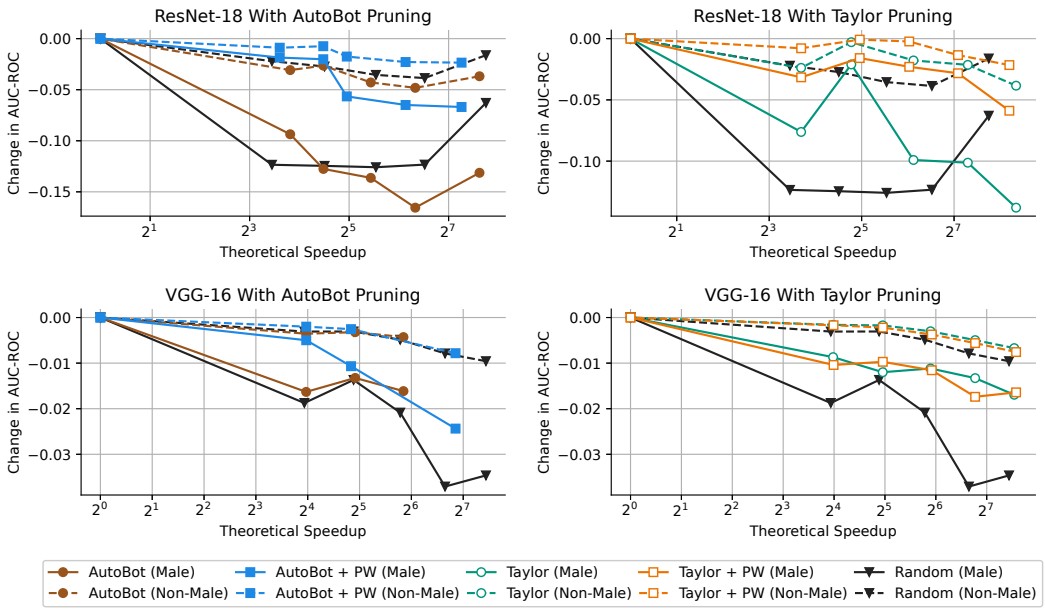

Figure 1: Mean pruning performance with Resnet-18 and VGG-16 models with CelebA dataset.

### 4.2.1 Pruning the CelebA Models

We trained a Resnet-18 [30] model and a VGG-16 [31] model for the CelebA task. The ROC-AUCs for the male and non-male subgroups of the Resnet-18 model were 0.9639 and 0.9794 respectively. The ROC-AUCs for the male and non-male subgroups of the VGG-16 model were 0.9679 and 0.9825 respectively. Both models were pruned using target theoretical speedups of 16, 32, 64, 128 and 256.

The change in ROC-AUC for all tested pruning methods for the Resnet-18 and VGG-16 models can be found in Figure 1. Some of the VGG-16 models pruned using the AutoBot, AutoBot with performance weighting and random methods always predicted a single class. These degenerate models were excluded from the figure. All methods were able to significantly reduce the size of both models, but most of the results without performance weighting exhibited divergent performance between the male and non-male subgroups as the theoretical speedup increases. Performance weighting was highly effective when pruning the Resnet-18 model for both the AutoBot and Taylor pruning methods. We see an increase in AUC-ROC at all tested theoretical speedups for both the male and non-male subgroups. The increase for the male subgroup is substantial and the subgroup AUC-ROC scores no longer diverage as the theoretical speedup increases.

We see similar improvements when performance weighting is applied to the AutoBot method for the VGG-16 model, however the improvements are only substantial at the lowest theoretical speedups. We do not see improvements when performance weighting is applied to the Taylor method. This is likely due to the method not having significantly divergent performance for the VGG-16 model.

### 4.2.2 Pruning the Fitzpatrick17k Models

We trained a Resnet-34 [30] model and a EfficientNet-V2 Medium [32] model for the Fitzpatrick17k task. The ROC-AUCs for the medium and dark subgroups of the Resnet-34 model were 0.8190 and 0.7329 respectively. The ROC-AUCs for the medium and dark subgroups of the EfficientNet model were 0.8516 and 0.7524 respectively.

Despite a bias against dark skin tones existing in the original models, we do not see divergent AUC-ROC scores as the theoretical speedup increases. The medium skin tone subgroup actually saw greater changes in AUC-ROC due to pruning. We only see slight benefits for using performance weighting with the Fitzpatrick17k models. Performance weighting increased slightly improved performance after pruning for the ResNet-34 model with AutoBot pruning and the EfficientNet model

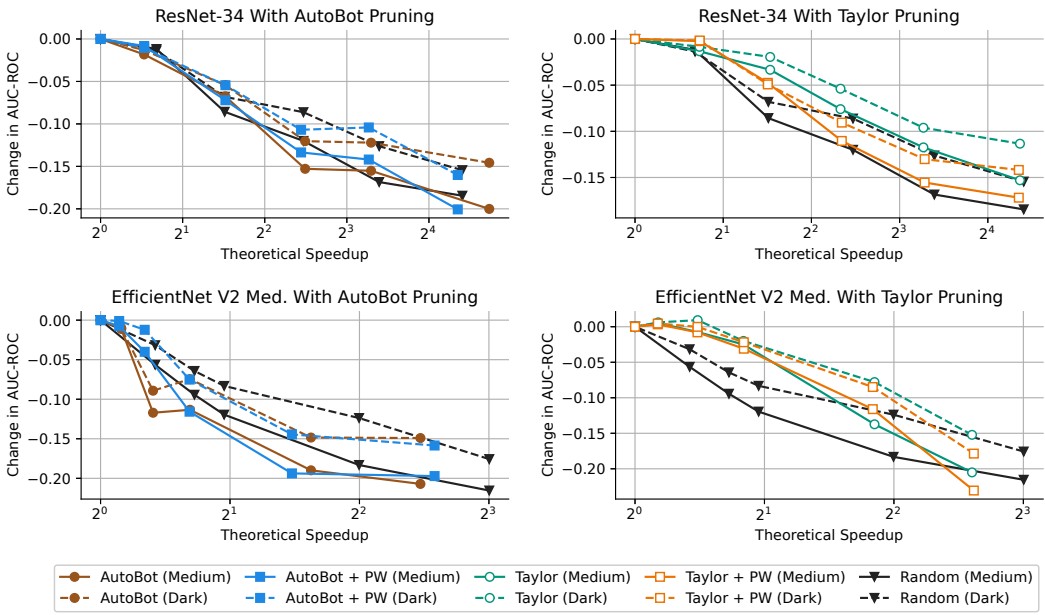

Figure 2: Mean pruning performance with Resnet-34 and EfficientNet V2 Med. models with Fitzpatrick17k dataset.

with AutoBot pruning at lower theoretical speedups. It had negligible or detrimental effects for Taylor pruning with both models.

These results indicate that performance weighting is not an appropriate solution for all datasets and models that exhibit bias. The lack of an increasing performance difference between subgroups may indicate that the pruning process was not introducing additional biases in the Fitzpatrick17k models. This is in contrast to the CelebA models for which the initial bias was small but grew due to pruning. Performance weighting may therefore only mitigate biases that are introduced from the pruning process. It will not rectify biases that exist in the model before pruning.

### 4.3 Conditions for Bias

From our results in Section 4.2, we can see that utilizing the PW loss is not necessary in all circumstances. The loss appeared to be more beneficial for models which saw increasing differences in performance between subgroups as the theoretical speedup increased.

To understand the properties of a dataset that would necessitate the use of the PW loss, we created three artificial datasets from the CelebA dataset by selected subsets of the training data. The first subset was formed using 3.41% of the available training data such that it was fully balanced, containing an equal number of male and non-male samples as well as an equal number of blonde and non-blonde samples. The second and third subsets were formed by adding additional samples to the first subset, altering the class or gender balance. The second subset contained an equal number of blonde and non-blonde samples, but five times as many non-male samples as there were male samples. The third subset contained an equal number of male and non-male samples, but five times as many non-blonde samples as there were blonde samples. The entire test set was used to evaluate all subsets.

A ResNet-18 model was trained using each subset. The AUC-ROCs for the male subgroup are 0.9562, 0.9479 and 0.9183 for the first, second and third subsets respectively. The AUC-ROCs for the non-male subgroup are 0.9713, 0.9732 and 0.9580 for the first second and third subsets respectively. The models were pruned using the AutoBot and Taylor methods using target theoretical speedups of 8, 32 and 128. The performance after pruning for these models can be found in Figure 3.

In the results using the fully balanced subset, we do see a divergence in subgroup performance for both methods, but the divergence is less than was seen when the full method was used. In the results with the additional non-male samples, we see an increase in performance for all model/method

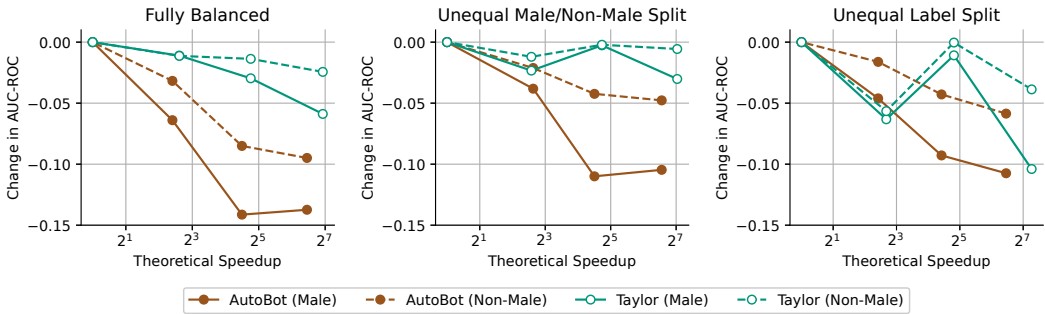

Figure 3: Pruning performance with ResNet-18 models trained on subsets of CelebA dataset with alternative class and gender balances.

combinations. For the AutoBot results, the increase is greater for non-male samples than it is for male samples. We once again see a common increase in performance when we look at the subset with the additional non-blonde samples. We do see additional instability in the Taylor results, but there are no clear findings with respect to differences in performance between subgroups. A greater decrease in performance was seen for male samples for all model/method combinations, including those that were trained on data with a balanced gender split. These results indicate that the dataset composition does influence the fairness of pruning results, but it does not fully explain it.

### 4.4 Ablation

To measure the effects of the components of the PW loss independently, we pruned our ResNet-18 CelebA model using the AutoBot method with only the corrected soft-labels and with only the weighting scheme described in equation 1. We applied the modifications to the only the pruning process, and to both the pruning and retraining processes.

The ablation results can be found in Figure 4. Both the modifications were more effective when applied to both the pruning and retraining process, indicating that simply modifying the process by which parameters are selected to be pruning is insufficient to mitigate the effects of bias. Furthermore, the effect of using corrected soft-labels was larger than the effects of using our proposed weighting scheme. While both changes boosted performance for the male subgroup when applied to both pruning and retraining, the effect of the corrected soft-labels was almost as large the effect of the full performance weighting method. The full method did demonstrate less bias with a target theoretical speedup of 16. Furthermore, as the AutoBot method already uses the outputs of the original model in its loss function, the improvement seen when the corrected soft-labels were only used for pruning can solely be attributed to the correction of the model outputs.

Unlike our proposed weighting scheme, the use of corrected soft-labels does not involve the selection of any parameters. In situations in which parameter selection is not possible, the use of corrected soft-labels may be a simple yet useful method for reducing the effects of algorithmic bias in pruning.

## 5   Conclusion

In this paper we demonstrate how model pruning can exacerbate biases in models and present the performance weighted loss function as a novel method for mitigating this effect. The performance weighted loss function is a simple modification that can be applied to any pruning method that uses the cross-entropy loss. Our experimental results indicate that while the performance weighted loss function does not recitify model biases, it can help prevent those biases from becoming exaggerated by the pruning process. The performance weighted loss function is a useful tool for practioners who seek to compress existing models without introducing new fairness concerns.

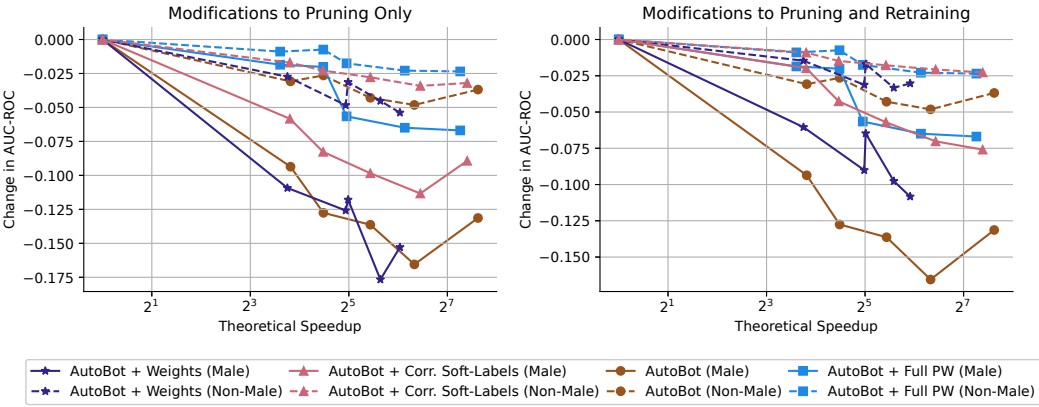

Figure 4: Pruning performance with ResNet-18 models with CelebA dataset when elements of PW loss are applied independently to the pruning process (left), and to the pruning process as well as the post-prune retraining process (right).

## Acknowledgements

This work was funded by the Natural Sciences and Engineering Research Council of Canada, the Ontario Graduate Scholarship and the Canada Research Chairs Program. We thank them for their support.

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

## A   Model Training and Pruning Parameters

To ensure transparency and enable reproducability, all parameters and procedures used to train, prune and retrain the models can be found below. All experiments were implemented using *PyTorch* 1.12.1 and *torchvision* 0.13.1 [27]. *PyTorch Lightning* 1.7.1 [33] was also used to train the models.

The ResNet-18 [30] CelebA model was trained for 20 epochs using the AdamW [34] optimizer with an initial learning rate of 0.0001 and a CosineAnnealingLR learning rate scheduler with $T_{max} = 20$ [35]. A batch size of 256 was used. The model was initialized using the provided ImageNet weights from *torchvision*. All parameters in layers except the final fully connected layer were frozen for the first 5 epochs after which they were unfrozen with a learning rate equal to 0.01 times the global learning rate. Early stopping was applied such that the parameters that achieved the lowest validation loss were saved after training.

The VGG-16 [31] CelebA model was trained for 10 epochs using the AdamW [34] optimizer with an initial learning rate of 0.0005 and a CosineAnnealingLR learning rate scheduler with $T_{max} = 10$ [35]. A batch size of 64 was used. The model was initialized using the provided ImageNet weights from *torchvision*. All parameters in layers except the final fully connected layer were optimized with a learning rate equal to 0.01 times the global learning rate. Early stopping was applied such that the parameters that achieved the lowest validation loss were saved after training.

The ResNet-34 [30] Fitzpatrick17k model was trained for 30 epochs using the AdamW [34] optimizer with an initial learning rate of 0.001 and a CosineAnnealingLR learning rate scheduler with $T_{max} = 30$ [35]. A batch size of 64 was used. The model was initialized using the provided ImageNet weights from *torchvision*. All parameters in layers except the final fully connected layer were frozen for the first 5 epochs after which they were unfrozen with a learning rate equal to 0.001 times the global learning rate.

The EfficientNet V2 Medium [32] Fitzpatrick17k model was trained for 30 epochs using the AdamW [34] optimizer with an initial learning rate of 0.001 and a CosineAnnealingLR learning rate scheduler with $T_{max} = 30$ [35]. A batch size of 32 was used. The model was initialized using the provided ImageNet weights from the *torchvision*. All parameters in layers except the final fully connected layer were frozen for the first 5 epochs after which they were unfrozen with a learning rate equal to 0.01 times the global learning rate.

The parameter values used for our AutoBot [18] implementation can be found in Table 1. $\beta_{AB}$ and $\gamma_{AB}$ refer to the parameters used by the AutoBot method to control the balance between the different terms of its loss function.

The parameter values used for our Taylor [14] implementation can be found in Table 2. $f_{\text{prune}}$ refers to the frequency of the pruning. That is, the number of batch iterations between the pruning of filters. $N_{\text{filters}}$ refers to the number of convolutional filters that are pruned in each pruning instance.

The parameter values that are used for our PW losses can be found in Table 3. Other parameters were not changed when the PW loss was introduced.

After pruning, all models were retrained using the AdamW [34] optimizer and CosineAnnealingLR learning rate scheduler with a $T_{max}$ value equal to the number of epochs. The parameter values used to retrain the models can be found in Table 4.

Table 1: Parameters used for AutoBot pruning method

| Dataset | Model | Learning Rate | Batch Size | Iters. | $\beta_{AB}$ | $\gamma_{AB}$ |
|---------|-------|---------------|------------|--------|--------------|---------------|
| CelebA | ResNet-18 | 0.85 | 64 | 200 | 2.7 | 0.1 |
| CelebA | VGG-16 | 1.81 | 64 | 250 | 3.07 | 0.18 |
| Fitzpatrick17k | ResNet-34 | 1.5 | 32 | 400 | 0.5 | 1 |
| Fitzpatrick17k | EfficientNet V2 Med. | 1.5 | 16 | 600 | 6.76 | 1.05 |

Table 2: Parameters used for Taylor pruning method

| Dataset | Model | Learning Rate | Batch Size | $f_{\text{prune}}$ | $N_{\text{filters}}$ |
|---------|-------|---------------|------------|--------------------|----------------------|
| CelebA | ResNet-18 | 0.01 | 64 | 5 | 1 |
| CelebA | VGG-16 | 0.01 | 64 | 5 | 1 |
| Fitzpatrick17k | ResNet-34 | 0.01 | 32 | 5 | 1 |
| Fitzpatrick17k | EfficientNet V2 Med. | 0.01 | 16 | 4 | 8 |

Table 3: Parameters used for PW loss

| Dataset | Model | Base Method | $\theta$ | $\gamma$ |
|---------|-------|-------------|----------|----------|
| CelebA | ResNet-18 | AutoBot | 0.3 | 1 |
| CelebA | ResNet-18 | Taylor | 0.8 | 0.5 |
| CelebA | VGG-16 | AutoBot | 0.75 | 3 |
| CelebA | VGG-16 | Taylor | 0.9 | 5 |
| Fitzpatrick17k | ResNet-34 | AutoBot | 0.8 | 2.5 |
| Fitzpatrick17k | ResNet-34 | Taylor | 0.95 | 3 |
| Fitzpatrick17k | EfficientNet V2 Med. | AutoBot | 0.8 | 2 |
| Fitzpatrick17k | EfficientNet V2 Med. | Taylor | 0.95 | 3 |

Table 4: Parameters used to retrain models

| Dataset | Model | Learning Rate | Batch Size | Duration |
|---------|-------|---------------|------------|----------|
| CelebA | ResNet-18 | 0.0001 | 256 | 30 epochs |
| CelebA | VGG-16 | 0.0005 | 64 | 10 epochs |
| Fitzpatrick17k | ResNet-34 | 0.0001 | 64 | 30 epochs |
| Fitzpatrick17k | EfficientNet V2 Med. | 0.00001 | 32 | 50 epochs |

# B  Detailed Results

For brevity, we only included figures displaying our results in the main body of this report. For transparency, our full results can be found here. Detailed results for all experiments performed in the main body of the paper can be found in Tables 5 through 10.

Table 5: Pruning performance mean ± standard deviation with ResNet-18 with CelebA dataset

| Pruning Method | FLOPS | Parameters | Accuracy | ROC-AUC | | |
| --- | --- | --- | --- | --- | --- | --- |
| | | | | All | Male | Non-Male |
| *Unpruned* | 1508.5 M | 11177.0 k | 0.9824 | 0.9546 | 0.9639 | 0.9795 |
| AutoBot | 107.1 M | 187.5 k | 0.9494 | 0.9378 | 0.8703 | 0.9487 |
| | ± 1.4 M | ± 16.6 k | ± 0.0066 | ± 0.002 | ± 0.0307 | ± 0.005 |
| AutoBot | 67.4 M | 811.7 k | 0.9511 | 0.9428 | 0.8364 | 0.953 |
| | ± 0.6 M | ± 35.8 k | ± 0.0068 | ± 0.0038 | ± 0.0346 | ± 0.0053 |
| AutoBot | 34.8 M | 545.4 k | 0.9364 | 0.9313 | 0.8276 | 0.9366 |
| | ± 1.7 M | ± 68.0 k | ± 0.0012 | ± 0.0033 | ± 0.0532 | ± 0.0041 |
| AutoBot | 18.7 M | 277.0 k | 0.9298 | 0.9299 | 0.7984 | 0.9313 |
| | ± 1.3 M | ± 16.6 k | ± 0.015 | ± 0.0054 | ± 0.054 | ± 0.0132 |
| AutoBot | 7.7 M | 72.4 k | 0.9419 | 0.9297 | 0.8326 | 0.9427 |
| | ± 0.5 M | ± 12.1 k | ± 0.0048 | ± 0.0038 | ± 0.0427 | ± 0.0055 |
| AutoBot + PW | 123.9 M | 345.7 k | 0.974 | 0.9442 | 0.9454 | 0.9706 |
| | ± 1.5 M | ± 29.2 k | ± 0.0013 | ± 0.0029 | ± 0.0042 | ± 0.0021 |
| AutoBot + PW | 67.6 M | 611.2 k | 0.975 | 0.9442 | 0.9438 | 0.9721 |
| | ± 0.6 M | ± 148.8 k | ± 0.002 | ± 0.0015 | ± 0.0064 | ± 0.0023 |
| AutoBot + PW | 48.7 M | 734.2 k | 0.9646 | 0.936 | 0.9074 | 0.962 |
| | ± 5.0 M | ± 47.7 k | ± 0.0039 | ± 0.0017 | ± 0.0123 | ± 0.0039 |
| AutoBot + PW | 21.5 M | 294.8 k | 0.9588 | 0.9317 | 0.899 | 0.9566 |
| | ± 1.3 M | ± 10.6 k | ± 0.0019 | ± 0.0022 | ± 0.0024 | ± 0.002 |
| AutoBot + PW | 9.8 M | 108.0 k | 0.9579 | 0.9282 | 0.897 | 0.956 |
| | ± 1.7 M | ± 16.6 k | ± 0.0019 | ± 0.0028 | ± 0.0079 | ± 0.0031 |
| Taylor | 116.8 M | 70.8 k | 0.9571 | 0.9445 | 0.8877 | 0.9556 |
| | ± 5.0 M | ± 2.2 k | ± 0.0128 | ± 0.001 | ± 0.05 | ± 0.01 |
| Taylor | 55.0 M | 25.8 k | 0.9785 | 0.9508 | 0.9429 | 0.9766 |
| | ± 2.4 M | ± 0.4 k | ± 0.0023 | ± 0.0037 | ± 0.0114 | ± 0.0018 |
| Taylor | 21.7 M | 9.8 k | 0.9593 | 0.9517 | 0.8648 | 0.9617 |
| | ± 1.9 M | ± 0.7 k | ± 0.0168 | ± 0.0012 | ± 0.074 | ± 0.0124 |
| Taylor | 9.6 M | 4.4 k | 0.9564 | 0.9476 | 0.8626 | 0.9581 |
| | ± 0.8 M | ± 0.4 k | ± 0.0197 | ± 0.001 | ± 0.0684 | ± 0.0156 |
| Taylor | 4.6 M | 1.9 k | 0.9388 | 0.9103 | 0.8259 | 0.9411 |
| | ± 0.9 M | ± 0.7 k | ± 0.0425 | ± 0.0378 | ± 0.1451 | ± 0.0346 |
| Taylor + PW | 116.7 M | 62.8 k | 0.9741 | 0.9488 | 0.9323 | 0.9716 |
| | ± 0.7 M | ± 1.9 k | ± 0.0031 | ± 0.0023 | ± 0.0087 | ± 0.0032 |
| Taylor + PW | 48.5 M | 21.9 k | 0.9805 | 0.9528 | 0.948 | 0.9787 |
| | ± 0.6 M | ± 0.6 k | ± 0.0007 | ± 0.0026 | ± 0.0046 | ± 0.0006 |
| Taylor + PW | 23.1 M | 9.6 k | 0.9788 | 0.9535 | 0.9409 | 0.9772 |
| | ± 2.2 M | ± 0.9 k | ± 0.0009 | ± 0.0024 | ± 0.0072 | ± 0.0011 |
| Taylor + PW | 11.1 M | 3.9 k | 0.9711 | 0.9403 | 0.9356 | 0.966 |
| | ± 0.3 M | ± 0.7 k | ± 0.0036 | ± 0.0044 | ± 0.0127 | ± 0.0074 |
| Taylor + PW | 5.1 M | 1.8 k | 0.9601 | 0.9229 | 0.9051 | 0.9579 |
| | ± 1.8 M | ± 0.2 k | ± 0.006 | ± 0.0146 | ± 0.0176 | ± 0.0059 |
| Random | 138.2 M | 1022.8 k | 0.9542 | 0.9478 | 0.8404 | 0.9574 |
| | ± 2.8 M | ± 41.3 k | ± 0.0038 | ± 0.0014 | ± 0.0236 | ± 0.0031 |
| Random | 66.4 M | 496.6 k | 0.9497 | 0.9448 | 0.8393 | 0.9522 |
| | ± 5.6 M | ± 30.6 k | ± 0.0058 | ± 0.0004 | ± 0.0281 | ± 0.0066 |
| Random | 32.4 M | 176.3 k | 0.9428 | 0.9406 | 0.838 | 0.944 |
| | ± 0.7 M | ± 19.7 k | ± 0.0013 | ± 0.0018 | ± 0.007 | ± 0.0015 |
| Random | 16.4 M | 120.9 k | 0.94 | 0.9342 | 0.8406 | 0.9409 |
| | ± 1.7 M | ± 5.6 k | ± 0.0059 | ± 0.0026 | ± 0.0125 | ± 0.0046 |
| Random | 7.0 M | 24.1 k | 0.9641 | 0.942 | 0.901 | 0.9633 |
| | ± 0.7 M | ± 10.9 k | ± 0.0113 | ± 0.004 | ± 0.0448 | ± 0.0092 |

Table 6: Pruning performance mean ± standard deviation with VGG-16 with CelebA dataset

| Pruning Method | FLOPS | Parameters | Accuracy | ROC-AUC | | |
| --- | --- | --- | --- | --- | --- | --- |
| | | | | All | Male | Non-Male |
| *Unpruned* | 11782.0 M | 134264.6 k | 0.9852 | 0.9586 | 0.9679 | 0.9826 |
| AutoBot | 751.9 M | 22104.4 k | 0.9813 | 0.9559 | 0.9516 | 0.979 |
| | ± 2.0 M | ± 343.0 k | ± 0.0002 | ± 0.0006 | ± 0.0016 | ± 0.0002 |
| AutoBot* | 390.5 M | 21966.0 k | 0.9817 | 0.9557 | 0.9547 | 0.9793 |
| | ± 2.6 M | ± 144.5 k | ± 0.0 | ± 0.0002 | ± 0.0017 | ± 0.0004 |
| AutoBot* | 204.7 M | 21889.2 k | 0.9807 | 0.9548 | 0.9518 | 0.9783 |
| | ± 0.6 M | ± 140.7 k | ± 0.0013 | ± 0.0012 | ± 0.0055 | ± 0.0011 |
| AutoBot + PW* | 751.2 M | 21609.1 k | 0.9832 | 0.9568 | 0.9629 | 0.9805 |
| | ± 1.2 M | ± 283.8 k | ± 0.0001 | ± 0.001 | ± 0.0016 | ± 0.0001 |
| AutoBot + PW** | 412.5 M | 21652.6 k | 0.9827 | 0.958 | 0.9572 | 0.98 |
| AutoBot + PW** | 102.0 M | 21086.5 k | 0.9775 | 0.9503 | 0.9436 | 0.9747 |
| Taylor | 746.1 M | 19832.1 k | 0.9833 | 0.9581 | 0.9592 | 0.9809 |
| | ± 3.2 M | ± 514.9 k | ± 0.0006 | ± 0.0006 | ± 0.0035 | ± 0.0003 |
| Taylor | 379.5 M | 19171.8 k | 0.9829 | 0.9582 | 0.9559 | 0.9808 |
| | ± 3.8 M | ± 421.7 k | ± 0.0002 | ± 0.0009 | ± 0.0016 | ± 0.0004 |
| Taylor | 197.9 M | 18949.7 k | 0.9821 | 0.9559 | 0.9568 | 0.9795 |
| | ± 3.2 M | ± 230.9 k | ± 0.0008 | ± 0.0015 | ± 0.0032 | ± 0.0009 |
| Taylor | 107.8 M | 18805.9 k | 0.9804 | 0.9542 | 0.9546 | 0.9776 |
| | ± 2.5 M | ± 1.2 k | ± 0.0011 | ± 0.0008 | ± 0.0041 | ± 0.001 |
| Taylor | 63.3 M | 18735.3 k | 0.9788 | 0.9524 | 0.951 | 0.9758 |
| | ± 1.1 M | ± 115.1 k | ± 0.0002 | ± 0.0004 | ± 0.0026 | ± 0.0007 |
| Taylor + PW | 741.4 M | 19416.6 k | 0.9831 | 0.958 | 0.9575 | 0.9809 |
| | ± 3.3 M | ± 467.6 k | ± 0.0002 | ± 0.001 | ± 0.0024 | ± 0.0003 |
| Taylor + PW | 379.1 M | 19099.7 k | 0.9827 | 0.9563 | 0.9582 | 0.9803 |
| | ± 3.3 M | ± 515.6 k | ± 0.0009 | ± 0.0012 | ± 0.002 | ± 0.0012 |
| Taylor + PW | 194.6 M | 18746.9 k | 0.9815 | 0.9549 | 0.9564 | 0.9788 |
| | ± 3.0 M | ± 817.5 k | ± 0.0009 | ± 0.0018 | ± 0.0024 | ± 0.001 |
| Taylor + PW | 107.8 M | 18403.9 k | 0.9797 | 0.9534 | 0.9505 | 0.9769 |
| | ± 0.4 M | ± 878.4 k | ± 0.0014 | ± 0.0019 | ± 0.0036 | ± 0.0014 |
| Taylor + PW | 61.7 M | 18132.0 k | 0.9782 | 0.9507 | 0.9515 | 0.975 |
| | ± 1.4 M | ± 418.7 k | ± 0.0006 | ± 0.0009 | ± 0.0042 | ± 0.0005 |
| Random* | 770.0 M | 43645.0 k | 0.9814 | 0.9564 | 0.9492 | 0.9795 |
| | ± 1.0 M | ± 773.7 k | ± 0.0004 | ± 0.0011 | ± 0.0002 | ± 0.0004 |
| Random | 398.7 M | 36418.8 k | 0.9819 | 0.9552 | 0.9542 | 0.9795 |
| | ± 0.6 M | ± 611.7 k | ± 0.0003 | ± 0.0007 | ± 0.0027 | ± 0.0004 |
| Random** | 213.7 M | 32481.6 k | 0.9799 | 0.9537 | 0.9471 | 0.9777 |
| Random* | 117.4 M | 26650.8 k | 0.9765 | 0.9508 | 0.9308 | 0.9747 |
| | ± 0.1 M | ± 146.0 k | ± 0.0002 | ± 0.0007 | ± 0.0012 | ± 0.0 |
| Random* | 67.9 M | 22757.6 k | 0.9753 | 0.9486 | 0.9333 | 0.973 |
| | ± 0.1 M | ± 422.0 k | ± 0.0001 | ± 0.0008 | ± 0.0039 | ± 0.0004 |

* indicates one of the three trials failed ** indicates two of the three trials failed

Table 7: Pruning performance mean ± standard deviation with ResNet-34 with Fitzpatrick17k dataset

| Pruning Method | FLOPS | Parameters | Accuracy | ROC-AUC | | |
| --- | --- | --- | --- | --- | --- | --- |
| | | | | All | Medium | Dark |
| *Unpruned* | 3682.0 M | 21286.2 k | 0.8023 | 0.7896 | 0.819 | 0.7375 |
| AutoBot | 2549.2 M | 9425.2 k | 0.7852 | 0.7852 | 0.8007 | 0.7244 |
| | ± 53.6 M | ± 178.6 k | ± 0.0114 | ± 0.0138 | ± 0.0134 | ± 0.007 |
| AutoBot | 1285.0 M | 4263.8 k | 0.7375 | 0.7535 | 0.7514 | 0.6832 |
| | ± 20.4 M | ± 358.1 k | ± 0.0043 | ± 0.0016 | ± 0.0038 | ± 0.0086 |
| AutoBot | 654.5 M | 1163.9 k | 0.6565 | 0.6792 | 0.6661 | 0.6172 |
| | ± 36.1 M | ± 282.1 k | ± 0.0129 | ± 0.0144 | ± 0.009 | ± 0.0272 |
| AutoBot | 375.1 M | 978.9 k | 0.653 | 0.6816 | 0.6638 | 0.6154 |
| | ± 26.5 M | ± 78.5 k | ± 0.0165 | ± 0.0157 | ± 0.019 | ± 0.0061 |
| AutoBot | 138.3 M | 24.0 k | 0.6127 | 0.7421 | 0.6189 | 0.5917 |
| | ± 6.5 M | ± 5.7 k | ± 0.0171 | ± 0.0111 | ± 0.018 | ± 0.0159 |
| AutoBot + PW | 2547.4 M | 8955.0 k | 0.7935 | 0.783 | 0.8107 | 0.7271 |
| | ± 48.2 M | ± 201.7 k | ± 0.0042 | ± 0.0049 | ± 0.0051 | ± 0.0007 |
| AutoBot + PW | 1283.0 M | 4078.3 k | 0.7333 | 0.7554 | 0.7468 | 0.6831 |
| | ± 90.3 M | ± 112.2 k | ± 0.0103 | ± 0.009 | ± 0.0104 | ± 0.0216 |
| AutoBot + PW | 677.3 M | 1670.9 k | 0.6736 | 0.7096 | 0.6854 | 0.6306 |
| | ± 39.6 M | ± 308.8 k | ± 0.0132 | ± 0.0131 | ± 0.0157 | ± 0.0084 |
| AutoBot + PW | 382.3 M | 1283.4 k | 0.6679 | 0.7099 | 0.677 | 0.6333 |
| | ± 12.8 M | ± 126.6 k | ± 0.0068 | ± 0.013 | ± 0.0065 | ± 0.0206 |
| AutoBot + PW | 180.4 M | 68.5 k | 0.6103 | 0.7261 | 0.6183 | 0.5774 |
| | ± 12.5 M | ± 27.4 k | ± 0.0127 | ± 0.0266 | ± 0.0135 | ± 0.0067 |
| Taylor | 2219.5 M | 6614.4 k | 0.7897 | 0.7831 | 0.8056 | 0.729 |
| | ± 12.1 M | ± 90.9 k | ± 0.0137 | ± 0.0144 | ± 0.012 | ± 0.0242 |
| Taylor | 1273.0 M | 2271.0 k | 0.7712 | 0.7744 | 0.7856 | 0.718 |
| | ± 14.3 M | ± 60.2 k | ± 0.0116 | ± 0.0078 | ± 0.0134 | ± 0.0126 |
| Taylor | 732.5 M | 879.8 k | 0.73 | 0.7492 | 0.743 | 0.6836 |
| | ± 7.1 M | ± 22.5 k | ± 0.0184 | ± 0.0073 | ± 0.0176 | ± 0.0232 |
| Taylor | 381.8 M | 333.0 k | 0.6877 | 0.7061 | 0.7015 | 0.6414 |
| | ± 6.9 M | ± 21.8 k | ± 0.0166 | ± 0.0117 | ± 0.018 | ± 0.015 |
| Taylor | 178.8 M | 127.9 k | 0.6565 | 0.6961 | 0.6659 | 0.6241 |
| | ± 7.8 M | ± 16.8 k | ± 0.0073 | ± 0.0194 | ± 0.0094 | ± 0.0099 |
| Taylor + PW | 2214.8 M | 6752.7 k | 0.8003 | 0.791 | 0.8166 | 0.7359 |
| | ± 8.6 M | ± 60.7 k | ± 0.003 | ± 0.0078 | ± 0.0026 | ± 0.0054 |
| Taylor + PW | 1296.7 M | 2406.0 k | 0.7534 | 0.7506 | 0.7715 | 0.6883 |
| | ± 26.5 M | ± 121.6 k | ± 0.0018 | ± 0.0109 | ± 0.0036 | ± 0.0145 |
| Taylor + PW | 724.9 M | 886.0 k | 0.6944 | 0.7088 | 0.7085 | 0.6469 |
| | ± 3.5 M | ± 7.0 k | ± 0.0129 | ± 0.0108 | ± 0.0112 | ± 0.0197 |
| Taylor + PW | 378.1 M | 340.0 k | 0.6509 | 0.682 | 0.6635 | 0.6073 |
| | ± 9.8 M | ± 18.2 k | ± 0.0169 | ± 0.0205 | ± 0.0147 | ± 0.0288 |
| Taylor + PW | 180.4 M | 142.1 k | 0.6363 | 0.6981 | 0.6471 | 0.5957 |
| | ± 3.3 M | ± 6.8 k | ± 0.0093 | ± 0.02 | ± 0.0109 | ± 0.0073 |
| Random | 2296.8 M | 13762.6 k | 0.79 | 0.7904 | 0.8071 | 0.7239 |
| | ± 33.0 M | ± 114.3 k | ± 0.0033 | ± 0.01 | ± 0.005 | ± 0.0086 |
| Random | 1291.3 M | 7771.6 k | 0.7191 | 0.7572 | 0.7332 | 0.6694 |
| | ± 35.8 M | ± 333.6 k | ± 0.0042 | ± 0.0068 | ± 0.0045 | ± 0.0026 |
| Random | 664.0 M | 4011.0 k | 0.6884 | 0.7255 | 0.6989 | 0.6513 |
| | ± 36.7 M | ± 254.9 k | ± 0.0125 | ± 0.0213 | ± 0.0141 | ± 0.0209 |
| Random | 350.3 M | 2122.7 k | 0.6419 | 0.7147 | 0.6505 | 0.6113 |
| | ± 10.1 M | ± 113.5 k | ± 0.0068 | ± 0.0151 | ± 0.0075 | ± 0.0027 |
| Random | 173.6 M | 925.2 k | 0.6236 | 0.6652 | 0.6344 | 0.583 |
| | ± 2.8 M | ± 86.2 k | ± 0.0184 | ± 0.0156 | ± 0.0182 | ± 0.0202 |

Table 8: Pruning performance mean ± standard deviation with EfficientNet V2 Med. with Fitzpatrick17k dataset

| Pruning Method | FLOPS | Parameters | Accuracy | ROC-AUC | | |
| --- | --- | --- | --- | --- | --- | --- |
| | | | | All | Medium | Dark |
| *Unpruned* | 5464.7 M | 52862.2 k | 0.831 | 0.8218 | 0.8516 | 0.7524 |
| AutoBot | 4892.1 M | 41883.3 k | 0.8202 | 0.8168 | 0.8405 | 0.7424 |
| | ± 4.0 M | ± 939.3 k | ± 0.0062 | ± 0.0086 | ± 0.0082 | ± 0.0053 |
| AutoBot | 4125.9 M | 41931.5 k | 0.7196 | 0.755 | 0.7345 | 0.6632 |
| | ± 254.6 M | ± 3481.1 k | ± 0.0794 | ± 0.038 | ± 0.0822 | ± 0.0709 |
| AutoBot | 3385.7 M | 42110.8 k | 0.7253 | 0.7577 | 0.7383 | 0.6779 |
| | ± 91.6 M | ± 523.8 k | ± 0.0292 | ± 0.0175 | ± 0.0323 | ± 0.0164 |
| AutoBot | 1771.0 M | 20118.8 k | 0.65 | 0.7041 | 0.6617 | 0.604 |
| | ± 94.8 M | ± 1568.1 k | ± 0.021 | ± 0.0192 | ± 0.0254 | ± 0.0196 |
| AutoBot | 985.7 M | 13211.3 k | 0.6357 | 0.7123 | 0.6446 | 0.6033 |
| | ± 33.3 M | ± 278.8 k | ± 0.0259 | ± 0.0101 | ± 0.0284 | ± 0.012 |
| AutoBot + PW | 4946.1 M | 41855.0 k | 0.8251 | 0.8164 | 0.8441 | 0.7513 |
| | ± 65.8 M | ± 896.3 k | ± 0.0024 | ± 0.0019 | ± 0.0028 | ± 0.001 |
| AutoBot + PW | 4314.5 M | 40312.4 k | 0.7969 | 0.7972 | 0.8116 | 0.7402 |
| | ± 214.6 M | ± 4882.1 k | ± 0.0353 | ± 0.018 | ± 0.0406 | ± 0.0157 |
| AutoBot + PW | 3393.3 M | 42535.4 k | 0.7235 | 0.7577 | 0.7359 | 0.6772 |
| | ± 94.2 M | ± 937.1 k | ± 0.0409 | ± 0.0296 | ± 0.0406 | ± 0.0403 |
| AutoBot + PW | 1961.2 M | 21341.5 k | 0.6474 | 0.7007 | 0.6579 | 0.6079 |
| | ± 74.7 M | ± 1960.4 k | ± 0.0177 | ± 0.0081 | ± 0.0204 | ± 0.0115 |
| AutoBot + PW | 913.7 M | 11856.2 k | 0.6411 | 0.7315 | 0.6544 | 0.5939 |
| | ± 161.7 M | ± 2435.1 k | ± 0.0094 | ± 0.0124 | ± 0.0115 | ± 0.0059 |
| Taylor | 4826.1 M | 33486.3 k | 0.8366 | 0.8277 | 0.8567 | 0.7586 |
| | ± 1.5 M | ± 46.3 k | ± 0.0029 | ± 0.0015 | ± 0.0039 | ± 0.0004 |
| Taylor | 3907.0 M | 17731.0 k | 0.8274 | 0.8168 | 0.8442 | 0.7617 |
| | ± 0.8 M | ± 33.4 k | ± 0.0018 | ± 0.0026 | ± 0.0011 | ± 0.0029 |
| Taylor | 3057.6 M | 8477.2 k | 0.8071 | 0.7909 | 0.8267 | 0.7324 |
| | ± 2.0 M | ± 61.2 k | ± 0.0048 | ± 0.0049 | ± 0.0073 | ± 0.0061 |
| Taylor | 1515.9 M | 1448.1 k | 0.705 | 0.696 | 0.7143 | 0.6745 |
| | ± 13.0 M | ± 13.3 k | ± 0.0495 | ± 0.0498 | ± 0.051 | ± 0.0404 |
| Taylor | 899.2 M | 668.7 k | 0.6369 | 0.7275 | 0.6466 | 0.6002 |
| | ± 21.1 M | ± 19.1 k | ± 0.0155 | ± 0.0028 | ± 0.0188 | ± 0.0069 |
| Taylor + PW | 4836.8 M | 33734.0 k | 0.8346 | 0.8275 | 0.855 | 0.757 |
| | ± 1.5 M | ± 55.8 k | ± 0.0068 | ± 0.0041 | ± 0.0062 | ± 0.0083 |
| Taylor + PW | 3911.8 M | 17826.1 k | 0.8247 | 0.809 | 0.8437 | 0.7523 |
| | ± 2.1 M | ± 21.9 k | ± 0.0019 | ± 0.0079 | ± 0.0028 | ± 0.0024 |
| Taylor + PW | 3051.9 M | 8429.1 k | 0.8021 | 0.7592 | 0.8207 | 0.7305 |
| | ± 4.4 M | ± 60.3 k | ± 0.0054 | ± 0.0149 | ± 0.0061 | ± 0.0082 |
| Taylor + PW | 1529.8 M | 1460.9 k | 0.7216 | 0.747 | 0.7356 | 0.6676 |
| | ± 8.6 M | ± 14.9 k | ± 0.0071 | ± 0.0136 | ± 0.0047 | ± 0.0137 |
| Taylor + PW | 890.6 M | 672.0 k | 0.6113 | 0.7539 | 0.621 | 0.5738 |
| | ± 12.8 M | ± 8.8 k | ± 0.0084 | ± 0.0016 | ± 0.0069 | ± 0.0184 |
| Random | 4078.8 M | 39661.8 k | 0.7794 | 0.7838 | 0.7952 | 0.7207 |
| | ± 157.9 M | ± 539.2 k | ± 0.0424 | ± 0.0238 | ± 0.0441 | ± 0.0351 |
| Random | 3305.1 M | 31406.7 k | 0.7427 | 0.7647 | 0.7573 | 0.6881 |
| | ± 6.9 M | ± 869.5 k | ± 0.0202 | ± 0.012 | ± 0.0202 | ± 0.0206 |
| Random | 2819.2 M | 26370.3 k | 0.7185 | 0.7451 | 0.7321 | 0.6692 |
| | ± 28.5 M | ± 515.0 k | ± 0.0053 | ± 0.0142 | ± 0.007 | ± 0.0041 |
| Random | 1369.1 M | 13060.2 k | 0.6602 | 0.7172 | 0.6685 | 0.6286 |
| | ± 33.3 M | ± 157.0 k | ± 0.0192 | ± 0.0098 | ± 0.0208 | ± 0.0123 |
| Random | 681.8 M | 6646.2 k | 0.6241 | 0.7162 | 0.6362 | 0.5767 |
| | ± 6.3 M | ± 44.0 k | ± 0.0153 | ± 0.0501 | ± 0.0142 | ± 0.0153 |

Table 9: Pruning performance mean $\pm$ standard deviation with ResNet-18 with subsets of CelebA

| Subset | Pruning Method | FLOPS | Parameters | ROC-AUC | |
|---|---|---|---|---|---|
| | | | | Male | Non-Male |
| Fully Balanced | *Unpruned* | 1508.5 M | 11177.0 k | 0.9562 | 0.9713 |
| Fully Balanced | AutoBot | 285.5 M | 497.9 k | 0.8922 | 0.9396 |
| | | $\pm$ 1.7 M | $\pm$ 86.7 k | $\pm$ 0.031 | $\pm$ 0.0048 |
| Fully Balanced | AutoBot | 66.9 M | 864.8 k | 0.8148 | 0.8863 |
| | | $\pm$ 1.5 M | $\pm$ 30.9 k | $\pm$ 0.0275 | $\pm$ 0.0212 |
| Fully Balanced | AutoBot | 17.2 M | 196.0 k | 0.8188 | 0.8764 |
| | | $\pm$ 3.0 M | $\pm$ 35.5 k | $\pm$ 0.0137 | $\pm$ 0.0337 |
| Fully Balanced | Taylor | 247.2 M | 192.7 k | 0.9451 | 0.9601 |
| | | $\pm$ 6.6 M | $\pm$ 15.5 k | $\pm$ 0.0026 | $\pm$ 0.0009 |
| Fully Balanced | Taylor | 55.7 M | 31.4 k | 0.9265 | 0.9576 |
| | | $\pm$ 1.8 M | $\pm$ 2.7 k | $\pm$ 0.0097 | $\pm$ 0.004 |
| Fully Balanced | Taylor | 12.4 M | 5.7 k | 0.8973 | 0.9469 |
| | | $\pm$ 0.4 M | $\pm$ 0.4 k | $\pm$ 0.0201 | $\pm$ 0.008 |
| Unequal Male/Non-Male Split | *Unpruned* | 1508.5 M | 11177.0 k | 0.9479 | 0.9732 |
| Unequal Male/Non-Male Split | AutoBot | 243.4 M | 1868.5 k | 0.9098 | 0.952 |
| | | $\pm$ 1.6 M | $\pm$ 80.0 k | $\pm$ 0.0102 | $\pm$ 0.0019 |
| Unequal Male/Non-Male Split | AutoBot | 66.5 M | 923.6 k | 0.838 | 0.9308 |
| | | $\pm$ 3.0 M | $\pm$ 16.3 k | $\pm$ 0.0226 | $\pm$ 0.007 |
| Unequal Male/Non-Male Split | AutoBot | 16.0 M | 174.6 k | 0.8433 | 0.9255 |
| | | $\pm$ 1.2 M | $\pm$ 14.6 k | $\pm$ 0.0209 | $\pm$ 0.0057 |
| Unequal Male/Non-Male Split | Taylor | 252.9 M | 205.8 k | 0.9246 | 0.9611 |
| | | $\pm$ 4.4 M | $\pm$ 20.1 k | $\pm$ 0.0086 | $\pm$ 0.0065 |
| Unequal Male/Non-Male Split | Taylor | 56.8 M | 30.8 k | 0.9453 | 0.971 |
| | | $\pm$ 1.8 M | $\pm$ 0.3 k | $\pm$ 0.0083 | $\pm$ 0.0021 |
| Unequal Male/Non-Male Split | Taylor | 11.5 M | 5.9 k | 0.9178 | 0.9675 |
| | | $\pm$ 1.8 M | $\pm$ 0.1 k | $\pm$ 0.0099 | $\pm$ 0.0045 |
| Unequal Label Split | *Unpruned* | 1508.5 M | 11177.0 k | 0.9183 | 0.958 |
| Unequal Label Split | AutoBot | 281.8 M | 723.8 k | 0.8722 | 0.9418 |
| | | $\pm$ 3.4 M | $\pm$ 38.7 k | $\pm$ 0.0117 | $\pm$ 0.0065 |
| Unequal Label Split | AutoBot | 70.4 M | 915.0 k | 0.8255 | 0.9151 |
| | | $\pm$ 4.9 M | $\pm$ 95.9 k | $\pm$ 0.052 | $\pm$ 0.0123 |
| Unequal Label Split | AutoBot | 17.1 M | 208.4 k | 0.8109 | 0.8994 |
| | | $\pm$ 1.1 M | $\pm$ 44.6 k | $\pm$ 0.0147 | $\pm$ 0.0102 |
| Unequal Label Split | Taylor | 235.8 M | 187.5 k | 0.8551 | 0.9015 |
| | | $\pm$ 3.1 M | $\pm$ 6.2 k | $\pm$ 0.0242 | $\pm$ 0.0189 |
| Unequal Label Split | Taylor | 53.7 M | 22.5 k | 0.9075 | 0.9576 |
| | | $\pm$ 4.9 M | $\pm$ 0.7 k | $\pm$ 0.057 | $\pm$ 0.0262 |
| Unequal Label Split | Taylor | 9.8 M | 4.1 k | 0.8143 | 0.9193 |
| | | $\pm$ 1.0 M | $\pm$ 0.5 k | $\pm$ 0.0758 | $\pm$ 0.037 |

Table 10: Pruning performance mean $\pm$ standard deviation with ResNet-18 with CelebA when elements of PW loss are applied independently

| Area of Modification | Pruning Method | FLOPS | Parameters | ROC-AUC | |
| --- | --- | --- | --- | --- | --- |
| | | | | Male | Non-Male |
| Pruning Only | AutoBot + Weights | 111.9 M $\pm$ 3.5 M | 188.2 k $\pm$ 59.2 k | 0.8547 $\pm$ 0.0354 | 0.9521 $\pm$ 0.0041 |
| Pruning Only | AutoBot + Weights | 47.6 M $\pm$ 6.2 M | 278.5 k $\pm$ 62.1 k | 0.846 $\pm$ 0.025 | 0.9481 $\pm$ 0.0089 |
| Pruning Only | AutoBot + Weights | 49.3 M $\pm$ 3.3 M | 729.4 k $\pm$ 41.3 k | 0.8381 $\pm$ 0.0071 | 0.9312 $\pm$ 0.0022 |
| Pruning Only | AutoBot + Weights | 23.0 M $\pm$ 12.4 M | 350.8 k $\pm$ 219.1 k | 0.811 $\pm$ 0.0327 | 0.9256 $\pm$ 0.0183 |
| Pruning Only | AutoBot + Weights | 30.2 M $\pm$ 1.9 M | 474.6 k $\pm$ 30.9 k | 0.7873 $\pm$ 0.0246 | 0.9343 $\pm$ 0.0019 |
| Pruning Only | AutoBot + Corr. Soft-Labels | 108.1 M $\pm$ 1.0 M | 215.0 k $\pm$ 50.5 k | 0.9056 $\pm$ 0.0438 | 0.9625 $\pm$ 0.0089 |
| Pruning Only | AutoBot + Corr. Soft-Labels | 67.7 M $\pm$ 2.8 M | 916.2 k $\pm$ 25.4 k | 0.8812 $\pm$ 0.0587 | 0.9566 $\pm$ 0.0115 |
| Pruning Only | AutoBot + Corr. Soft-Labels | 34.9 M $\pm$ 2.4 M | 476.0 k $\pm$ 33.7 k | 0.8655 $\pm$ 0.0471 | 0.9517 $\pm$ 0.0115 |
| Pruning Only | AutoBot + Corr. Soft-Labels | 17.2 M $\pm$ 1.4 M | 199.0 k $\pm$ 11.0 k | 0.8506 $\pm$ 0.0513 | 0.9454 $\pm$ 0.0163 |
| Pruning Only | AutoBot + Corr. Soft-Labels | 8.9 M $\pm$ 1.1 M | 84.7 k $\pm$ 8.5 k | 0.8746 $\pm$ 0.0202 | 0.9475 $\pm$ 0.011 |
| Pruning and Retraining | AutoBot + Weights | 112.2 M $\pm$ 2.7 M | 194.8 k $\pm$ 47.8 k | 0.9035 $\pm$ 0.058 | 0.9649 $\pm$ 0.0142 |
| Pruning and Retraining | AutoBot + Weights | 46.7 M $\pm$ 4.9 M | 255.4 k $\pm$ 50.8 k | 0.8992 $\pm$ 0.0605 | 0.9626 $\pm$ 0.0168 |
| Pruning and Retraining | AutoBot + Weights | 47.7 M $\pm$ 2.9 M | 730.3 k $\pm$ 29.5 k | 0.8738 $\pm$ 0.04 | 0.9483 $\pm$ 0.019 |
| Pruning and Retraining | AutoBot + Weights | 31.5 M $\pm$ 12.2 M | 486.7 k $\pm$ 205.6 k | 0.8663 $\pm$ 0.064 | 0.9462 $\pm$ 0.0254 |
| Pruning and Retraining | AutoBot + Weights | 25.0 M $\pm$ 9.5 M | 387.2 k $\pm$ 167.9 k | 0.8556 $\pm$ 0.0765 | 0.9492 $\pm$ 0.0164 |
| Pruning and Retraining | AutoBot + Corr. Soft-Labels | 108.0 M $\pm$ 1.3 M | 222.1 k $\pm$ 57.6 k | 0.9441 $\pm$ 0.0025 | 0.9705 $\pm$ 0.0007 |
| Pruning and Retraining | AutoBot + Corr. Soft-Labels | 68.3 M $\pm$ 4.0 M | 928.8 k $\pm$ 16.8 k | 0.9213 $\pm$ 0.0022 | 0.9647 $\pm$ 0.0007 |
| Pruning and Retraining | AutoBot + Corr. Soft-Labels | 35.2 M $\pm$ 2.9 M | 474.5 k $\pm$ 19.8 k | 0.9069 $\pm$ 0.0079 | 0.9617 $\pm$ 0.0031 |
| Pruning and Retraining | AutoBot + Corr. Soft-Labels | 17.5 M $\pm$ 0.2 M | 199.9 k $\pm$ 10.5 k | 0.8937 $\pm$ 0.0114 | 0.9588 $\pm$ 0.0026 |
| Pruning and Retraining | AutoBot + Corr. Soft-Labels | 9.0 M $\pm$ 1.6 M | 85.3 k $\pm$ 11.7 k | 0.888 $\pm$ 0.016 | 0.9568 $\pm$ 0.0063 |

