# OpenReview forum: "A Fair Loss Function for Network Pruning"
_NeurIPS.cc/2022/Workshop/TSRML — TSRML2022_

### Official Review · Reviewer_iRzT · 2022-10-14
**A Fairness Loss Term for Pruning Criteria for Neural Networks**

**Overall Rating:** 7

**Summary:**

This paper studies the exacerbation of demographic biases in neural network models due to pruning. To avoid disparities in performance due to pruning, a new loss term is proposed with which a pruning criterion can be easily augmented. The efficacy of the approach is demonstrated on CelebA and Fitzpatrick17k benchmarks.

**Strengths:**

* The exact proposed combination of techniques has not been explored in the literature.

* The problem addressed in the paper is practically relevant: model compression via pruning may exacerbate biases, and the proposed approach mitigates this effect.

* The experiments are pretty comprehensive: two datasets and four neural network architectures are considered. In addition, an ablation study is performed to investigate the role of individual steps/loss terms in the pruning procedure.

**Weaknesses:**

* The paper is sometimes vague regarding the exact technical contributions. For example, in Section 3.2, the performance-weighted loss function is described. This loss function has been studied extensively in the prior literature and is known as the focal loss (Lin et al., 2017). Moreover, one work has even proposed using this loss for debiasing (Mahabdi & Henderson, 2019). These points must be made clear, and the previous work must be adequately acknowledged and referenced.

* Similarly, there have been several other approaches to pruning that incorporate fairness considerations, e.g. works by Paganini (2020) and Marcinkevics et al. (2022). It would be good to at least mention and discuss the differences with these closely related techniques.

* Currently, there is no ablation to disentangle the effect of pruning and retraining. It would be interesting to see how pruning performs "without* retraining.

* I find it concerning that the trials producing degenerate models were discarded. It would be helpful to at least report how many there were.

* Figures could be simplified: instead of plotting two performance traces (males vs nonmales), one could plot the difference in AUC-ROC across the two groups of the sensitive attribute.

### References

Lin, T. Y., Goyal, P., Girshick, R., He, K., & Dollár, P. (2017). Focal loss for dense object detection. In *Proceedings of the IEEE international conference on computer vision* (pp. 2980-2988).

Mahabadi, R. K., & Henderson, J. (2019). Simple but effective techniques to reduce dataset biases. URL: https://openreview.net/forum?id=SJlCK1rYwB

Paganini, M. (2020). Prune responsibly. *arXiv:2009.09936*.

Marcinkevičs, R., Ozkan, E., & Vogt, J. E. (2022). Debiasing Deep Chest X-Ray Classifiers using Intra-and Post-processing Methods. *arXiv:2208.00781*.


**Overall Recommendation:**

The paper deals with a relevant and practical problem: how to avoid the exacerbation of biases due to the model compression by pruning? The proposed method is interesting as a novel combination of previously proposed elements and, in practice, is effective at reducing bias exacerbation across various architectures. In summary, I believe this paper is a good fit for the current workshop. Further improvements could be made by articulating the contribution more clearly and referencing relevant related work.

**Review Confidence:**

4: The reviewer is confident but not absolutely certain that the evaluation is correct

---

### Official Review · Reviewer_L6yh · 2022-10-21
**Simple Yet Nice Contribution**

**Overall Rating:** 8

**Summary:**

The paper proposes a neat loss function to utilize pruning as a way to improve fairness.
There has been works that have identified that pruning can amplify the biases in the model.
On the other hand this work augments a loss function that can "boost the fairness" while pruning.
Minor changes to the cross-entropy loss function is all, but experiments show significant improvements.
The paper also contains ample amount of empirical data to support its case.
One possible improvements may be more analytical evidences to support the approach.


**Strengths:**

* Very simple yet effective approach that is well supported by both empirical results and the writing.

**Weaknesses:**

* Analytical evidence to why it works

**Overall Recommendation:**

The work makes a very simple yet neat contribution that improves fairness with pruning.
The empirical results support the claim very well. the paper is well written.This paper should be in the program.

**Review Confidence:**

4: The reviewer is confident but not absolutely certain that the evaluation is correct

---

### Decision · Program_Chairs · 2022-10-23

**Decision:**

Accept

**Comment:**

Great work connecting fairness and pruning with sufficient empirical results.